# Mechanically Reconfigurable, Beam-Scanning Reflectarray and Transmitarray Antennas: A Review

**Mirhamed Mirmozafari \*,†**, **Zongtang Zhang †**, **Meng Gao †**, **Jiahao Zhao †**, **Mohammad Mahdi Honari †**, **John H. Booske** and **Nader Behdad**

Department of Electrical and Computer Engineering, University of Wisconsin-Madison, Madison, WI 53706, USA; zongtang.zhang@wisc.edu (Z.Z.); mgao45@wisc.edu (M.G.); jzhao335@wisc.edu (J.Z.); honarikalate@wisc.edu (M.M.H.); jhbooske@wisc.edu (J.H.B.); behdad@wisc.edu (N.B.)
\* Correspondence: mirmozafari@wisc.edu
† These authors contributed equally to this work.

**Abstract:** We review mechanically reconfigurable reflectarray (RA) and transmitarray (TA) antennas. We categorize the proposed approaches into three major groups followed by a hybrid category that is made up of a combination of the three major approaches. We discuss the examples in each category and compare their performance metrics including aperture efficiency, gain, bandwidth and scanning range and resolution. We also identify opportunities to build upon or extend these demonstrated approaches to realize further advances in antenna performance.

**Keywords:** reconfigurable antennas; reflectarray antennas; transmitarray antennas; mechanically beam-scanning; scanning range





## 1. Introduction

The advent of modern applications requiring antenna arrays with flexible scanning capabilities has turned agile beam-scanning antenna design into an active area of research [1]. Some of these diverse applications include automotive radars, weather observations, air surveillance [2] and satellite communication (SatCom) [3]. The stringent requirements of these applications limit the antenna options to the following: (i) conventional mechanically-rotated dish reflectors, (ii) active phased array antennas and (iii) reflectarray (RA) and transmitarray (TA) antennas. The mechanically-rotated dish antenna is simple but it is neither compact nor sufficiently fast. Therefore, it is not considered an appropriate solution, especially for applications requiring fast scanning [1]. By contrast, active phased arrays offer flexible and fast beam scanning [2]. However, they need a designated Transmit/Receive (T/R) module for each antenna element, rendering them relatively complex solutions. Moreover, their radiated power levels are rather limited, making them only suitable for applications requiring low to moderate transmitting power. RA and TA antennas are the midway solutions between dish reflectors and active phased arrays. They have limited scanning capabilities compared to active phased arrays. However, using a single high-power amplifier connected to a feed antenna as an illuminating source and free-space as the medium to combine outgoing radiations from constituent elements provide RAs and TAs some advantages over active phased array antennas. These advantages have motivated many applications to consider RAs or TAs as their best option, especially those seeking simpler scanning scenarios, high-power and less complexity. Among the previously cited applications for agile beam scanning antenna development, SatCom has recently begun to adopt various forms of RAs and TAs [3].

Since TAs and RAs have similar properties and applications, we simplify our discussion by focusing on RAs in the rest of this introductory section. Despite the latent potentials for reconfigurablity, most RA-related publications have focused on static forms with fixed beams. This has begun to change over the last decade [4,5].

In order to discuss tunability for beam-scanning in RAs, Figure 1 illustrates the geometry of a RA including an illuminating feed antenna and an aperture of reflecting elements. To point the reflected beam in a given direction, one needs to impose a particular distribution of re-radiated (i.e., reflected) field phases across the array elements. The phase ($\Phi(x_i, y_i)$) of the re-radiated field from each element consists of two components: (i) the spatial delay due to the electrical distance ($k_0 R_i$) between the phase center of the feed antenna and the position of the element and (ii) the intrinsic reflection phase of the element ($\Phi_R(x_i, y_i)$). These two components are the first and the second terms of the equation below, respectively:

$$\Phi(x_i, y_i) = -k_0 R_i + \Phi_R(x_i, y_i), \tag{1}$$

where $k_0$ represents the free-space wavenumber and $(x_i, y_i)$ shows the location of the element in the array. One may obtain a beam-scannable RA by adjusting either or both of the two phase contributors. The same discussion holds for TAs with the difference that the reflection phases of elements are replaced by the transmission phases.

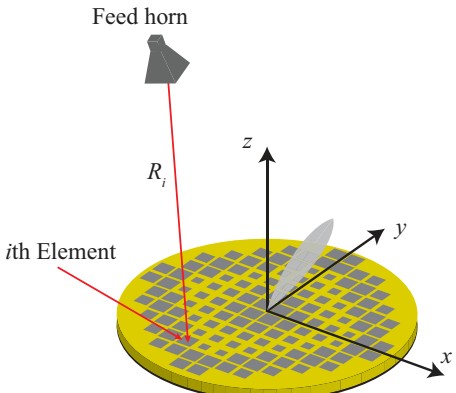

**Figure 1.** The geometry of a reflectarray antenna consisting of a feed antenna and a single planar aperture consisting of numerous spatial phase shifters.

The second term in (1) represents the element-level phase control that many reconfigurable RA research studies have attempted to achieve electronically [6–8]. Electronic adjustment of the constituting elements' reflection phases have been studied using pin diodes, varactor diodes and micro-electromechanical systems (MEMS). These electronically reconfigurable RAs have fast response times but encounter two major challenges. The first challenge is the amplitude loss of the element. This loss, typically ranging between 0.5 and 2 dB, is attributed to the presence of the phase electronic switching/tuning devices in the unit cell design. The other challenge is the phase quantization error, especially in RAs using switching-type elements. For a one-bit RA having two discrete phase states, the quantization error results in about 3 dB gain loss [9].

Alternatively, one can mechanically adjust the phase distribution of a RA. This can be conducted either by changing the relative distance of the feed antenna to the array elements on the aperture or by mechanically repositioning (tuning) the constituent elements. Generally, this mechanical tuning has a slower response time than the electronic reconfiguration due to limits on the speed of mechanical actuators. On the other hand, mechanical tuning often does not disturb unit-cell design. This minimizes a major contributor to element loss that manifests in electronically reconfigurable RAs. Actuators and micromotors are mostly integrated beneath the reflector so as to not block the RA's aperture. Finally, mechanically reconfigurable unit cells tend to have a wider range of re-radiated phase change, significantly reducing the phase quantization error.

In this paper, we review the state-of-the-art of mechanically reconfigurable beam-scanning RAs and TAs. We categorize the mechanically reconfigurable architectures into four categories: (i) element movement, (ii) array movement, (iii) feed movement and (iv) a hybrid category that represents combinations of the first three categories. We discuss

the advantages and disadvantages and typical performance metrics of each approach, including illustrative examples. We note that some of the approaches, especially element movement approaches, rely on the performance of actuators. Such actuators can be designed for various applications ranging from precision micron control of fragile optical components to the displacement of large masses [1]. This presents opportunities for the further development of mechanically-reconfigurable agile beam scanning RAs and TAs.

## 2. Element Movement

A mechanically reconfigurable RA can be realized by creating reconfigurable constituting elements. This requires a dedicated mechanical mechanism to each element and a method for distributed control. For large arrays with many elements, this can result in a complex overall system. On the other hand, individually tunable elements mostly offer full 0–360° reflected phase compensation. Therefore, this is a promising approach for applications requiring fine scanning resolution over a broad scanning range. The proposed methods in the literature to mechanically reconfigure elements in a RA can be categorized into two groups: (i) reconfigurable resonant elements and (ii) reconfigurable non-resonant elements.

### 2.1. Reconfigurable Resonant Elements
#### 2.1.1. Adjusting Dimensions

A common approach to locally control the re-radiated phase of individual elements in a RA is to adjust the lateral and height dimensions of resonant elements [1,10–12]. The incident electromagnetic field creates a standing wave between the resonant elements and the ground plane reflector [10]. Each resonant element has an equivalent reactance that varies with its dimensions. The interaction between the reactance and the standing wave causes the incident wave to be re-radiated with a phase shift. The resonant elements are tuned close to their resonant frequencies, resulting in limited frequency bandwidths. Crossed dipoles [10] and patch antennas [11] have been investigated as mechanically reconfigurable resonant elements. The dimensions of a rectangular patch antenna, for example, were finely tuned close to its resonant frequency to achieve different reflected phases [11]. A 350° phase difference was reported by adopting patches with varying size. The effective length of a patch can also be adjusted by changing the height of the patch over the ground plane [12]. This is accomplished by increasing or decreasing the fringing fields of elements, providing a mechanically reconfigurable RA with fixed size patches. Patch height displacement is less effective at changing the reflected phase than length adjustment. For example, a reradiated phase tuning range of only 197° was reported in [12].

Modifying the resonant features of a patch antenna can be conducted by adjusting its dimensions or by changing its excitation properties. Reference [1] proposes single-size patch antennas in a RA with coupling slots of different lengths to achieve different re-radiated phases. Three slots of varying sizes are laterally replaced beneath fixed patch elements of the RA to scan the beam to broadside and ±30°, providing the RA with limited reconfigurable capability. Alternatively, the relative gap distance of coupling slots to the patch and the ground plane can be tuned to shift the phase of the re-radiated fields [9]. This is shown in Figure 2, where the authors investigated a fixed square patch with a height-tunable coupling slot. This is an example of a mechanically reconfigurable RA where an actuator was integrated with each element and was located beneath the ground plane reflector. A plastic pedestal extends from the actuator and connects to the slotted patch. This arrangement has significantly modified the re-radiated phase range by as much as 324° by finely adjusting the coupling slots between the ground plane and the rectangular patch. The performance metrics of this RA are summarized in Table 1 and compared with representative RAs of other categories. These illustrative antenna examples represent the state-of-the-art of each approach and are built upon the constant progress and proposed concepts prior to them.

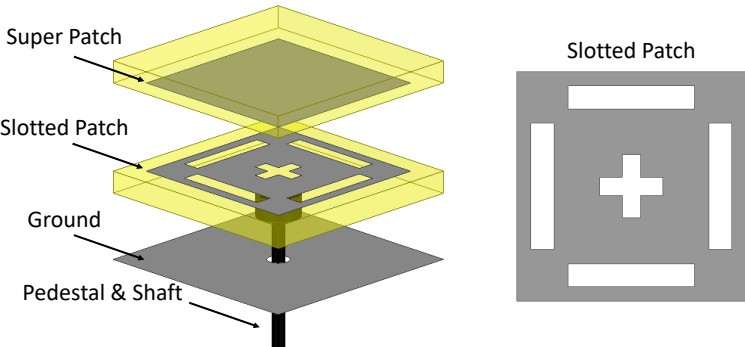

**Figure 2.** Height-tunable patch antenna unit cell of the RA in [9] with adjustable middle layer.

**Table 1.** Comparative analysis of mechanically reconfigurable RAs using reconfigurable elements.

| References | Methodology | Operating Frequency * (GHz)-Bandwidth | 3 dB Beam Scanning Range | Aperture Efficiency |
|---|---|---|---|---|
| [9] | Resonant elements—dimensions adjustment | 4.7–4.9, (4.7%) | ±60° | 48.6% |
| [4] | Resonant elements—rotating elements | 7.5–9, (17%) | ±60° | 51.8% |
| [13] | Non-resonant elements | 90–110, (20%) | NA | 50.1% |

* Determined based on 1 dB gain bandwidth.

### 2.1.2. Rotating Elements

The concept of phase variation using rotating elements was first demonstrated in a radio telescope where mechanically rotatable spiral antennas were used as feed elements for a parabolic reflector [14]. Building upon this concept, the authors of [15] investigated a linear array of patch antennas and demonstrated the scanning capability by mechanically rotating the elements in the array. The drawback of this approach is its need for rotational symmetry, thus, limiting this technique to circularly polarized (CP) RAs. The interaction between CP elements of a RA and the incident wave and the resulting phase shift is thoroughly discussed in [16]. Simply speaking, the incident and the reflected signals will have the same type of circular polarization if the element can reflect the incident wave with the same phase (reflection coefficient = +1) in one direction and with reversed phase (reflection coefficient = −1) in the orthogonal direction. The reflected wave is phase shifted by $2\alpha$ if the element of the RA is rotated by $\alpha$ degrees. Using this technique, one can achieve full phase compensation (360° phase shift) by rotating the CP element 180° around its axis. This could be a fruitful area for further research, integrating micromotors beneath the RA to create mechanically reconfigurable RAs with favorable features. Some of the related research are discussed below as illustrative examples.

Figure 3a shows a RA consisting of a circular patch having a center and two diametrically opposed shorting pins [17]. The fundamental $TM_{11}$ mode is excited beneath the circular patch antenna and phase shifting of the incident CP wave can be achieved by mechanically rotating the element. Reference [16] discusses the phase-shifting capabilities of different resonant elements such as a single rotating dipole, multiple dipoles and rotating stubs. Similar to the works discussed in Section 2.1.1, the RAs consisting of such elements have limited frequency bandwidths as they rely on the resonant features of the elements. The conventional techniques to broaden the frequency bandwidths of the elements, such as stacked patch antennas and tightly coupled elements, can consequently broaden the RA's frequency response. For example, a recently proposed RA, shown in Figure 3b, uses multiple concentric resonators to broaden the element's frequency bandwidth [4]. This is

also an example of a mechanically reconfigurable RA which includes a micromotor beneath each individual element. Full phase compensation is achieved by rotating the elements.

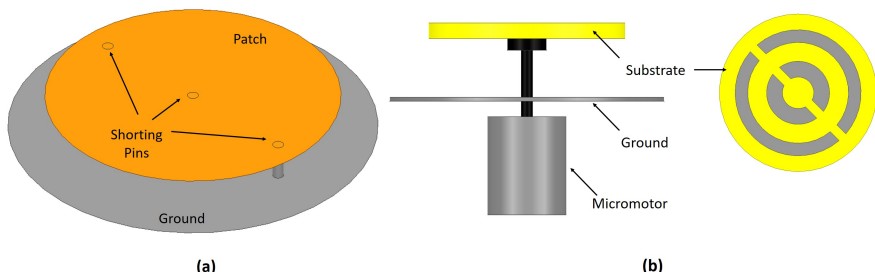

**Figure 3.** Illustrative reconfigurable resonant-type unit cells using rotating elements. (**a**) RA including a shorted circular disc. (**b**) Multi-resonator unit cell integrated with a micromotor beneath the reflector.

### 2.2. Reconfigurable Non-Resonant Elements

A different approach to introduce a phase shift to an incident wave on a reflector is to allow the incident wave to pass through a retarding medium. This can be a simple dielectric medium with a different refractive index than that of air. Building upon this concept, an RA populated by blocks of variable height dielectrics backed by a ground plane reflector was proposed in [18]. Full phase compensation can be achieved by selecting the proper height of each element. Based on this concept, an RA was 3D printed and characterized in [19]. The favorable advantages of RAs in this category over those using resonant elements are their wide frequency bandwidths, simplicity and low mutual couplings between elements. These dielectric-based RAs, although simple, have limited aperture efficiency mainly due to dielectric loss, which typically increases with increasing frequency. A 100 GHz example of this type of RA offers only a 14% aperture efficiency [13]. In order to address this issue, an all-metal RA was proposed in [13], replacing the dielectric blocks with metallic elements. Figure 4 illustrates the concept of this all-metal RA. Different phase shifts are achieved by varying the relative heights of the reflecting surface unit cells. The authors reported full 360° phase shift at 100 GHz by changing the metallic block's height with a resolution of 0.03 mm. Using low-loss full-metal elements has improved the aperture efficiency of the RA to more than 50%, much larger than other RA designs at these high frequencies. The performance metrics of the antenna discussed in [13] are provided in Table 1. The collective features of a metal-only RA render it a promising candidate for high-gain applications at submillimeter or THz bands. The actuators discussed in the previous category can be employed to vary the height of the metal blocks, creating a mechanically reconfigurable RA. This is a promising research area for future investigations.

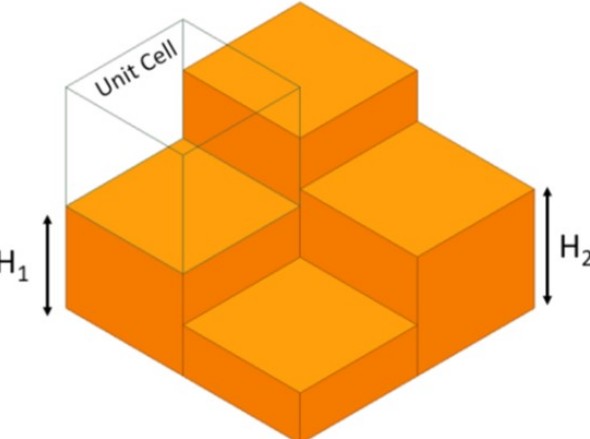

**Figure 4.** Reconfigurable non-resonant-type unit cell using metal blocks of variable heights.

## 3. Array Movement

In Sections 2.1 and 2.2, various mechanical RA designs deploying micro-motors or actuators to move or rotate each unit cell have been demonstrated to independently control the phase responses of individual unit cells. However, implementing individual control for the array elements results in increased cost and complexity, especially for arrays with many elements. This section reviews some works on mechanically reconfigurable TAs and RAs using array movement approaches, which only need a few actuators to implement the mechanical reconfiguration.

Array movement approaches that shift or rotate the whole array to scan the beam have been shown to provide a good compromise between practicality and performance. Most mechanical beam-steering systems using array movement methods only need one or two actuators. Therefore, mechanical beam-steerable phased arrays using array movement methods provide less complex and less expensive beam-steerable antenna options for applications that do not require high speed beam scanning and arbitrary beam-shaping capabilities. Mechanical beam-steering systems using array movement methods can be categorized into three main groups: (i) Risley-prism-based antennas, (ii) translatable or rotatable TA or RA antennas and (iii) tiltable ground plane RA antennas.

### 3.1. Risley-Prism-Based Antennas

Recently, mechanically beam-steerable antennas based on the Risley prism have attracted attention due to their simplicity and good beam scanning performance [20–26]. In the optical regime, a Risley-prism system typically consists of a pair of rotatable, wedge-shaped prisms used in conjunction with a well-collimated beam of light (e.g., a laser beam) [27,28]. The beam can be steered when the prisms are rotated with respect to one another. Later, the Risley prism concept was extended to the microwave regime. Different configurations of Risley-prism-based beam-steering systems were proposed and implemented [20–26]. Figure 5 shows three different configurations of Risley-prism-based beam-steerable antenna systems used at microwave frequencies [20–24,26]. Figure 5a shows the architecture of the first reported beam-steerable antenna that employed the Risely prism concept. Here, the two wedge-shaped prisms used in optical systems were replaced by a pair of planar, metasurface and effective prisms that were illuminated by a feed antenna. The spherical wave radiated by the feed antenna is collimated by the lens placed below the two flat prisms. The relative rotational displacement of the two flat prisms, Prism #1 and Prism #2, provides beam steering in both azimuth and elevation in the upper hemisphere of the outgoing aperture. This design configuration has been demonstrated at Ka-band with a $\pm 75°$ scan range [20]. The structure shown in Figure 5b combines the collimating lens with the bottom flat prism. In this manner, the single structure not only collimates the spherical wave radiated by the feed antenna but also provides the corresponding phase gradient needed from the first prism for beam steering. This architecture of a Risley-prism-based beam steering antenna has been used in [21,22]. Reference [21] presented a high-power-capable beam steering lens antenna working at X-band with a $\pm 20°$ beam scan range. In [22], a planar Risley-prism-based system was reported to generate a quasi-nondiffractive beam at Ka-band with a $\pm 54°$ beam scan range. Due to the use of the horn antenna that must be placed at some distance from the input aperture of the collimating lens/first prism, these approaches still have a large overall profile [20–22]. Such a high-profile is problematic in applications that require a compact geometry. Therefore, this high overall thickness motivated the development of the low-profile Risley-prism-based beam-steerable antenna architecture shown in Figure 5c. Here, the horn antenna was replaced by a planar antenna or an array to achieve a low-profile beam steering system [23,24]. In [23], a Fabry–Perot resonator antenna was used to feed the two prisms to steer the main beam with $\pm 51°$ range. This antenna system provides a peak gain of 19.4 dBi with a corresponding aperture efficiency of 19.2%. In [24], a dual-linear-polarized continuous transverse stub array was implemented to illuminate the two prisms and achieved a $\pm 40°$ beam scan range with an aperture efficiency of 17.8%.

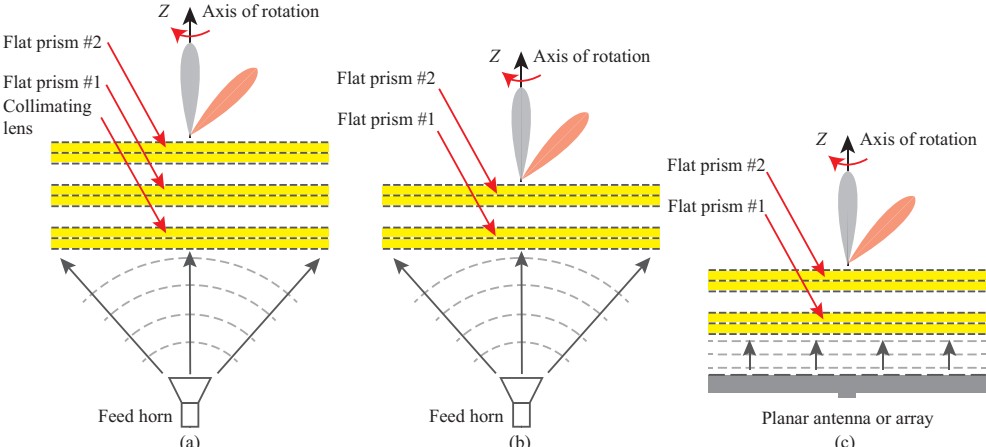

**Figure 5.** Different configurations of the Risley-prism-based beam-steerable antenna system reported in the literature. (**a**) A system employing a feed horn, a collimating lens and two flat prisms (used in [20]). (**b**) The system in part (**a**) in which the collimating lens are integrated with the first prism (used in [20–22,26]). (**c**) The system in part (**a**) in which the horn antenna is replaced with a planar antenna or antenna array placed in close proximity to the two flat prisms (used in [23,24]). In such a system, a collimating lens is not needed.

### 3.2. Translatable or Rotatable TA or RA Antennas

Two-dimensional beam steering capability can be achieved by only moving or rotating one lens antenna. Figure 6 shows the working principle of the antenna system and the lens mechanical steering setup. Lens in-plane translation with respect to the feed antenna provides beam steering in the elevation plane, while rotation of the lens antenna provides full azimuth coverage. This concept has been reported and implemented for wide angle (0–50°) elevation scanning of high-gain beams with full 360° azimuth sweep with a simple mechanical setup at Ka-band [29].

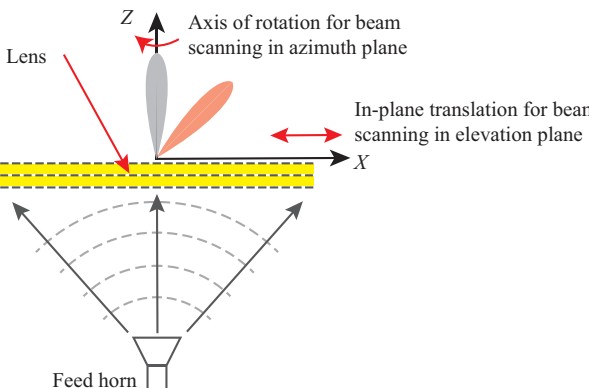

**Figure 6.** Schematic view of the lens mounting structure.

A mechanically reconfigurable reflector was reported in [30] based on a resonant textured ground plane, also known as a high-impedance surface. The tunable reflector consisted of a pair of printed circuit boards. When the two boards were placed against one another, the reflector functions as an array of tiny resonant cavities. By moving or rotating the upper array board, each individual tiny resonator was tuned to have a different reflected phase response. This resulted in a different phase gradient for the mechanically reconfigurable reflector by which the main beam was scanned. The presented structure was demonstrated to steer a reflected beam by ±38° at 3.1 GHz with a physical motion of only $\lambda/100$, where $\lambda$ is the free-space wavelength. A beam-steerable RA antenna designed at millimeter wave frequency band is presented in [31]. This RA aperture employs circular rings as its constituting phase shifters. The main beam of the proposed array can be steered

by tilting the entire RA panel at different angles from $+30°$ to $-30°$. The required tilt angles could be obtained by using a single stepper motor, which offers a much less complex implementation as well as lower power consumption compared to the arrays which require motors for each individual element of the apertures.

The mechanical tuning approach can also be applied to RAs for achieving polarization reconfiguration. Reference [32] reported a wideband and polarization-reconfigurable RA antenna for 5G millimeter wave wireless communications. The proposed 2 bit unit cell consists of a U-shaped dielectric stub with a cuboid-shaped air void and a ground plane. By simply rotating the reflective panel mechanically, the proposed RA antenna can switch between linear polarization (LP), left-hand circular polarization (LHCP) and right-hand circular polarization (RHCP) modes. This mechanical RA antenna has demonstrated a 3 dB gain bandwidth of 37.5%, 34.4% and 37.5% for RHCP, LHCP and LP modes, respectively.

### 3.3. Tiltable Ground Plane RA Antennas

Instead of tilting the whole RA, the work in [33] only tilts its ground plane for simplification. An X-band beam-steerable nonuniform RA antenna was reported, which achieved beam steering by using small movements of a large ground plane. The proposed RA exhibited a maximum $±7.5°$ beam scanning from the antenna broadside with only $±0.05\lambda$ ground tilting. This work also shows improved performance in terms of the half-power beamwidth (HPBW) and side lobe level, which were $9°$ and 20 dB, respectively. A new reconfigurable RA concept was studied in [34]. It involved only four mechanical actuators to change the orientation of an RF flexible ground plane to control the reflected phase. The ground plane was a flexible membrane that was deformable by actuators. Four actuators were distributed regularly along the membrane to control its vertical displacement. The magnitude of deformation was studied and the RF accuracy of the global system was optimized. The maximum slope of the membrane in this work was shown to be only 4.43%. This deformation slope is much lower than the required threshold of 15%, which is the acceptable stress in existing materials. A canonical 1D array providing three different shaped beams was designed and full-wave simulations were used to validate this preliminary design. Reference [35] reported the design of a reconfigurable high impedance surface (HIS) integrated with four commercial piezoelectric actuators. Piezoelectric actuators can convert an electrical signal into a precisely controlled physical displacement [36], which was translated to dynamic control of the reflection phase. The simulated and experimental results demonstrated a continuously tunable phase shift over $200°$ at about 60 GHz.

Another new approach using the concept of piezoelectric actuators to perform mechanical beam steering of a RA was presented in [37]. The proposed technique exploited collective control over macro-scale movements of a ground plane to achieve 2D beam steering. By tilting the ground plane underneath the entire aperture, the phase shift gradient created over the RA was modified. Changing the direction and slope of this phase shift gradient allowed for achieving 2D beam steering using only one control parameter (i.e., tilt vector of the ground plane). Experiments demonstrated that $0.05\lambda_0$ mechanical movements of the ground plane can steer the radiated beam direction in the $±10°$ range. They further demonstrated that this beam scanning range could be significantly enhanced to $±30°$ when the ground plane was segmented. The illustrations of the proposed mechanical reconfigurable RAs are shown in Figure 7 [37]. Figure 7a shows the configuration of the proposed beam-steerable RA that moves a large ground plane. Figure 7b illustrates an implementation form of Figure 7a, where four piezoelectric actuators are employed to control the height or the tilt angle of the ground plane. Figure 7c shows a conceptual configuration of the RA with a segmented ground plane, which allows for a wider beam scanning range. The performance metrics of this RA are summarized in Table 2 and compared with other representative mechanical beam-steerable antennas that employ array movement methods.

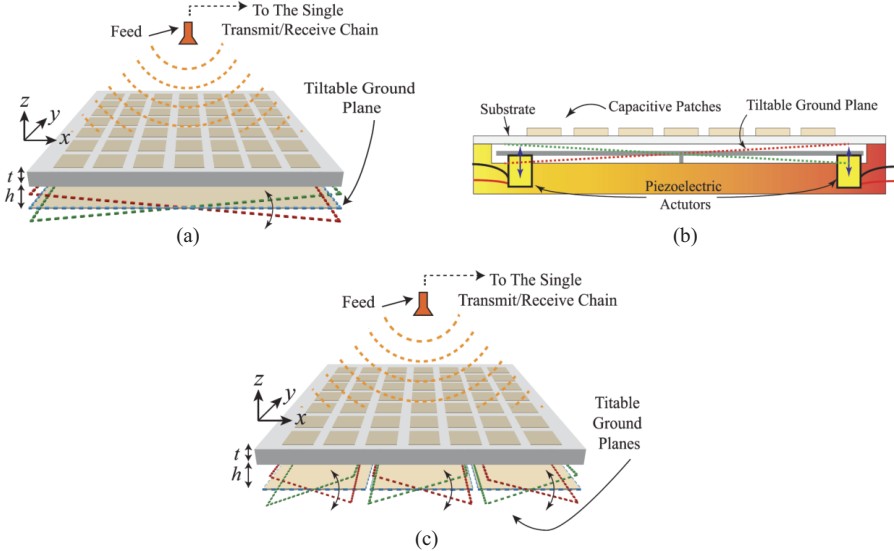

**Figure 7.** Illustrative typologies of mechanical beam-steerable reflectarray by moving/tilting the ground plane. (**a**) Reflectarray topology with tilting the single ground plane. (**b**) Piezoelectric actuators implemented prototype. (**c**) Topology of the reflectarray with a segmented ground plane. (Reprinted with permission from [37]).

**Table 2.** Comparative analysis of mechanically reconfigurable beam-steerable antennas using array movement approach.

| References | Methodology | Operating Frequency (GHz) | Beam Scanning Range | Aperture Efficiency |
|:---:|:---:|:---:|:---:|:---:|
| [20] | Risley-prism-based beam-steerable antenna | 30.0 | ±75° | 21.7% |
| [23] | Risley-prism-based beam-steerable antenna | 11.0 | ±51° | 19.2% |
| [29] | Moving and/or rotating a lens | 30.0 | ±50° | 20.5 |
| [37] | Titling the ground plane of RA | 9.5 | ±30° | NA |

### 4. Feed Movement

Moving the array (discussed in Section 3) requires fewer actuators, which simplifies mechanical reconfiguration. However, the implementation of actuators controlled by external circuits still renders the antenna system complicated. Moving the feed antenna further simplifies the realization of mechanically reconfigurable TA and RA antennas.

Feed displacement enables beam scanning by mechanically moving the phase center of the feed antenna over an aperture of elements along a displacement path, typically a focal arc, as shown in Figure 8. Moving the feed antenna creates spatial phase delay between the feed antenna and array aperture, which results in an adjustment of the phase center of the aperture. This generates the required phase distribution to direct the beam towards the desired direction. Feed displacement methods only rely on the mechanical movement of the feed antenna, thereby providing a simple configuration, low-cost implementation and low-power consumption.

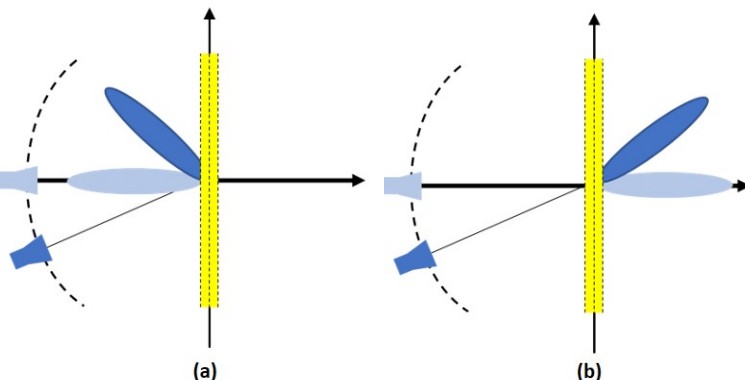

**Figure 8.** Feed displacement technique in (**a**) reflectarrays and (**b**) transmitarray.

With feed displacement, beam scanning performance, including scan coverage, scan speed and scan resolution [5], is determined by several factors including the aperture properties of the RA/TA, the focal arc position and the feed system characteristic. Typically, aperture properties and focal-distance-to-aperture-diameter (F/D ratio) determine the scan coverage, while the feed system's properties determine scan speed and resolution. Our discussion in this section emphasizes the impact of the array aperture and the focal position and does not focus on the feed system. Feed displacement techniques can be categorized into two general groups: (i) feed displacement on fully illuminated apertures and (ii) feed displacement on partially illuminated apertures.

### 4.1. Feed Displacement on Fully Illuminated Apertures

Beam scanning with antenna arrays using feed displacement have been extensively studied in [5,38–49,49–57] for RAs and in [58–69] for TAs. By convention, in fully illuminated RAs, the amplitude of the feed illumination at the aperture rim is less than −10 dB.

#### 4.1.1. Arrays with Single Focal Point

In arrays with a single focal point, the phase distribution on the aperture of RA/TA is designed to have a parabolic distribution in order to provide a collimated radiation beam with high gain [38,59]. It was shown in [39] that the lateral feed displacement scanning range of single focus configurations is restricted to a few beamwidths. This is due to the aberration introduced by the defocused phase center of the feed antenna and is improved by increasing F/D ratios, similar to parabolic dish reflector antennas. Furthermore, larger feed displacements introduce significant phase errors, especially for smaller values of the F/D ratio, since the reflection phases of the unit cells of the antenna array are functions of the angle of incidence. The beam scanning by lateral feed displacement for RAs with extremely short focal distances was investigated in [40] by using metasurface structures named metamirrors. Simulations and experiments demonstrated that the scanning capability of the metamirrors, despite having subwavelength focal distances, is comparable to RAs with focal distances of about several wavelengths. This is due to their deep subwavelength constituent elements which enables them to confine incident energy to extremely small focal distances. However, these RAs have high side lobe levels compared to RAs with larger focal distances. Although RAs with lateral feed displacement are easier to fabricate, displacement along a circular arc or spiral has demonstrated better performance [5]. Reference [58] proposed a metamaterial-based TA with a circular focal arc as a solution to obtain wide scanning coverage. Reference [58] also investigated the effects of the F/D ratio on the power distribution of the lens. A large F/D ratio yields a high scan resolution, low defocusing and a narrow scan coverage. In contrast, an F/D = 0.5 results in a very low aperture efficiency that is caused by low element illumination efficiency and a large taper effect. An FSS-based multibeam lens was developed in [70] by using multiple feeds placed on the

focal plane of the lens antenna. This is in contrast to conventional feed movements along the focal arc, offering the possibility to implement a complete planar feed system.

Single reflectors with parabolic phase distributions were investigated in early works on RAs [41]. In these RAs, the increase in spatial delay for scan angles far from broadside limited the scan range. Many attempts have been made to increase the scan range of RAs with feed displacements [42–53]. A multibeam phase matching method was studied in [42] to design wide-angle beam-scanning RAs. It was shown that optimizing the beam deviation factors and reference phase can minimize the total phase error. The RA unit cells were optimized to match the desired phase shift at multiple beam scanning angles. Measurement results showed that for scan angles from $-45°$ to $+45°$, the gain only drops 1.7 dB, which is a significant improvement in beam scanning performance compared to the conventional single-focal point RAs.

### 4.1.2. Arrays with Multi Focal points

The spatial phase error will increase when the feed antenna's phase center deviates from the focal point of the lens or RA, resulting in degradation of antenna gain and beam scanning performance. Each element on a RA/TA can provide a different phase delay independently, rendering it possible to use RAs/TAs to mimic conventional multifocal antennas by dynamically changing the phase shift on each element. Inspired by multifocal lenses and reflector antennas [44–46], a multifocal design approach was adopted for beam scanning of RAs [47,48] and TAs [65–68]. In the synthesis of multifocal RAs/TAs, it is required to find a compromise in the phase shift on each array element since they need to adjust the phase shift for multiple focal points. However, the phase error resulting from this method will slightly degrade the antenna gain at the initial scan angle, which is generally at broadside for single focal systems. Inspired by the enhanced scanning performance of bifocal reflector antennas [46,49], multifocal RA systems were reported to increase the scan range in numerous studies [49–53]. A bifocal RA antenna based on a dual-offset reflector configuration was investigated in [50]. While the beam scanning performance was considerably improved, high complexity and fabrication cost render this configuration undesirable. In [49], an RA with multiple focal points was designed that showed enhanced scanning performance compared to a single focal RA. Figure 9a shows the schematic of a bifocal RA system, where the aperture phase distribution needs to compensate for two different feed positions and beam directions. For a symmetric feed configuration and by using the average phase of both feed beams for each unit cell, it can be proved that the direction of the main beam of the RA will depend on the feed offset. Figure 9b shows the scanned gain patterns of a single-focal and a bifocal RA presented in [49]. As shown, the bifocal design provides better scan range in comparison with the single focal design. Moreover, while bifocal RAs provide a larger scan range, they have higher side lobe levels, which may be critical in some applications such as radar systems. This is probably due to the high dissimilarity in the spatial delay of the two feeds for larger scan angles and, therefore, causes higher phase errors while averaging the phases of the two feeds. By using pattern synthesis optimization and quadrufocal phase distribution for a single reflector, Reference [49] reported a wide beam scanning RA with scan coverage of 60°, aperture efficiency of 40% and side lobe level better than 15 dB. However, this improved performance compared to the single-focal parabolic-type design is achieved at the cost of lower aperture efficiency. The aperture efficiency of the quadrufocal RA reported in [49] is 15% lower than that of the single focal RA. A similar method was also used in designing a TA reported in [65]. In this design, four focal points were divided into two groups and located on the focal arcs in the E-plane and H-plane, respectively. This achieved a $\pm30°$ scanning coverage in both E-plane and H-plane with directivity variation less than 1 dB in the Ka-band. Optimization algorithms were used to balance both beam-scanning coverage and directivity and to maintain a stable directivity over the entire scanning coverage [69]. Conventionally, the phase distribution on a RA/TA is calculated by averaging the phase compensation of both feeds [65,66]. However, simply

averaging spatial phase delay from multiple focal points will cause significant phase error and radiation degradation since illumination from different focal point on array unit cells has non-negligible intensity difference [65,66]. In order to further reduce this phase error, a new formula in [67] was proposed to calculate the phase distribution on each unit cell. Instead of simply averaging the phase compensation of two feeds, the bifocal function is defined as a weighted average of the phase compensations. The values of weights were determined by the illumination intensity on each unit cell, giving greater weights to higher intensities. Reference [67] shows the superior scanning performance of bifocal designs compared to that of the unifocal designs of the same size. The side lobe level of bifocal design is $-13.8$ dB, while that of the unifocal design is $-8.1$ dB. Moreover, the bifocal design provides less than 3 dB scan loss, which is significantly less than 6 dB scan loss of the unifocal design over the similar scan range.

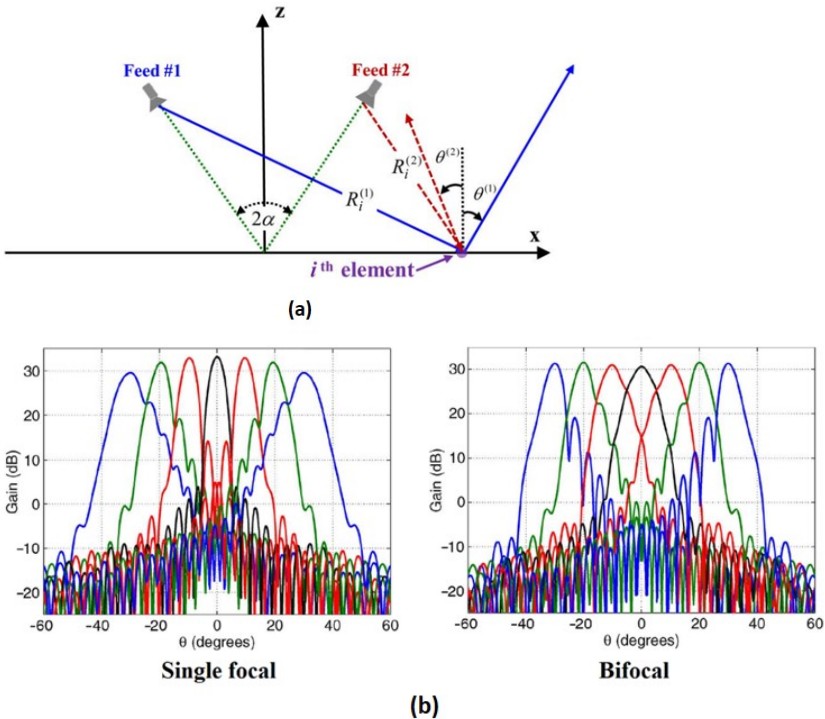

**Figure 9.** (**a**) Schematic of bifocal reflectarray antenna. (**b**) Scanned gain patterns of single and bifocal reflectarray designs (Reprinted with permission from [49]).

### 4.2. Feed Displacement on Partially Illuminated Apertures

There is a tradeoff between low phase aberration and high scan coverage when it comes to choosing the F/D ratio. Although some techniques were introduced to increase the scan range of RA, achieving a wide scan coverage is still difficult when using fully illuminated apertures. One method to address this issue is to use bigger reflectors for which different portions are used in order to create beam collimations toward different angles [54]. Indeed, using structures such as spheres or the parabolic torus [57] may increase the scan coverage. This is easily realizable in RAs since the phase of each unit cell can be simply controlled.

Benefitting from its spherical symmetry, the spherical reflector can provide a wide scan coverage by partially illuminating different parts of a perfectly symmetrical reflector surface. Therefore, spherical reflectors are suitable candidates for high-gain applications with wide-angle beam scanning [55]. The phase distribution over the aperture of a planar RA can be designed to be equivalent to that of a spherical reflector. An RA with a spherical phase distribution was studied in [56] and the optimum feed position was obtained by analyzing the phase error. By illuminating only a small part of the aperture for each beam

angle, the effect of spherical aberrations decreased. In this manner, an RA was designed demonstrating a gain of 30 dB, elevation coverage of 30° and an illuminated aperture size of 15λ for 2D beam scanning. Due to having lower phase aberrations, parabolic torus reflectors can achieve better scan ranges compared to their spherical counterparts. An offset parabolic-torus phase RA was designed in [57], where the offset-fed structure removed the feed blockage. Using this technique, an RA was designed at the Ka-band with a scan coverage of 40°. The performance metrics of some state-of-the-art RAs and TAs using feed movement are summarized in Table 3.

**Table 3.** Comparative analysis of reconfigurable RAs using feed movement.

| References | Methodology | Operating Frequency Bandwidth (GHz)-1 dB Gain | 3 dB Beam Scanning Range | Aperture Efficiency |
|---|---|---|---|---|
| [42] | Fully illuminated single focal array-reflectarray | 11.6–12.7 (9.1 %) | ±55° | 43.1% |
| [58] | Fully illuminated single focal array-transmitarray | NA | ±27° | 24.5% |
| [49] | Fully illuminated multifocal array-reflectarray | 31.6–33.2 (4.75 %) | ±30° | 40% |
| [56] | Partially illuminated array-reflectarray | NA | ±30° | 4.7% |

## 5. Hybrid Mechanical Movement Approach

A combination of mechanical movement techniques in element, array and feed levels can be used to scan the beam of RAs/TAs [71–76]. Reference [71] reported a folded RA antenna, which is capable of scanning its beam in two dimensions. This folded RA switched between seven feed beams in the E-plane while mechanically tilting the backside reflector for H-plane beam scanning. The proposed folded RA demonstrated a beam scanning performance of 16° in the E-plane and 9° in the H-plane at 76.5 GHz. A folded RA was proposed in [73] and consisted of a focusing array with a polarization rotating structure, a feed antenna and polarizing grids. The feed antenna and the focusing-and-polarization-rotating array were integrated within a single structure. By tilting this integrated structure, the phase center of the aperture was displaced and the antenna beam was steered at a rate twice the tilting angle of the integrated structure. The structure can scan the beam over ±10° from broadside without any significant degradation.

Another approach for combining feed displacement and array rotation was proposed in [74] which realized 1D ±70° beam scan coverage. The phase matching and mode superposition synthesis methods were combined to synthesize an ultra-wide RA scan range with excellent beam scanning performance. Specifically, the RA achieved a measured gain of 26.2 dBi at broadside at 12 GHz, corresponding to an aperture efficiency of 42.5%. The measured gain variations of 3.1 dB and 4.9 dB were reported for the scan ranges of ±60° and ±70°, respectively, demonstrating the ultra-wide scan coverage capability of this method. The TA antenna reported in [75] was implemented for Ka-band satellite-on-the-move terminals. Figure 10 illustrates how the beam-steering in the elevation plane was obtained by in-plane translation of the feed beam, while full azimuth coverage was obtained by rotating the whole TA. The developed prototype has a measured gain of 27 dBi at 30 GHz with a scan loss of 3 dB for an elevation beam-scanning range between 18° and 53°. Moreover, this antenna prototype has a weight of less than 500 grams and this configuration potentially requires very low DC power to move, which is an attractive feature for a user terminal. By using a similar mechanically reconfigurable mechanism, the same author also presented two offset dual-band transmit arrays in [76] possessing

23–29 dBi gain at 20 GHz(Rx) and 30 GHz(Tx), with an elevation beam scanning range of 0–50° and with a scan loss of less than 3.6 dB and F/D < 1.

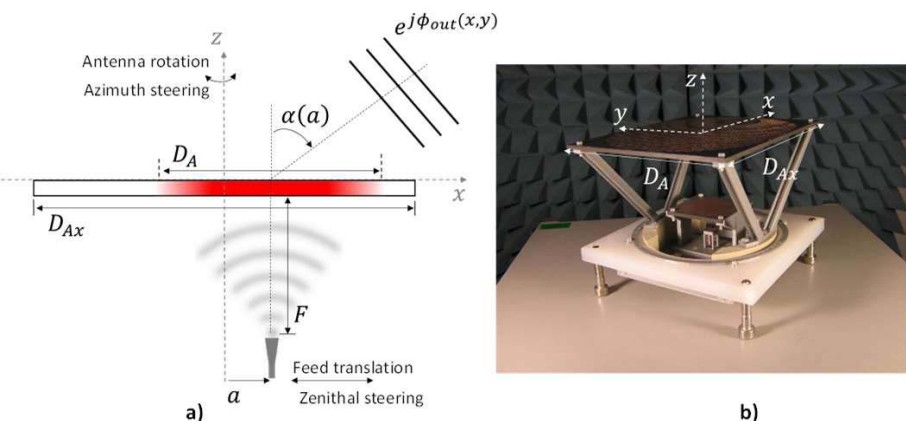

**Figure 10.** (**a**) Working principle of the hybrid beam scanning antenna. (**b**) Fabricated prototype. (Reprinted with permission from [76]).

## 6. Comparison and Discussion

In this section, we perform a comparison of the different approaches discussed to achieve mechanical reconfigurability. Table 4 compares the scanning capabilities, cost and complexity and power consumption of the three major approaches discussed herein. These parameters are rated good, fair and poor with green, yellow and red colors, respectively. By scanning capabilities, we refer to the scanning resolution, range and loss of each approach. As observed, the element movement approach offers better scanning capabilities as it permits adjustments at the element scale. This requires dedicating a micromotor to each element, rendering the array and the controller unit more complex and increasing power consumption. Moreover, the scanning capabilities of such arrays depend on the adopted micromotors features. Specifically, the adjustment speed and the size of the micromotors limit the scanning agility and the integration capability of the array, respectively. The impact of micromotor size on integration explains why most of the reported RAs using integrated actuators were designed at relatively low frequencies, i.e., C-band and S-band [4,9]. Nevertheless, the favorable scanning capabilities of this approach still render it appealing to many applications such as microwave radar and communication systems [77]. Furthermore, the recent advancement of miniaturized actuators and MEMS, which appear in a myriad of forms and sizes, introduces opportunities for designing reconfigurable RAs at higher frequencies with agile scanning capabilities. It is also possible to sacrifice part of the scanning capabilities of the arrays by using subarray movement and, in turn, reducing complexity and power consumption. By allocating a micromotor to a group of adjacent elements (one micromotor per subarray), one may obtain a simpler reconfigurable RA by giving up the full phase compensation of each element. This can also relax the size constraint of the actuators.

**Table 4.** Comparative analysis of reconfigurable approaches discussed in this paper.

| Method | Scanning Capabilities | Frequency Bandwidth | Cost and Complexity | Power Consumption |
|---|---|---|---|---|
| Element Movement | ✓ | ✓ | ✗ | ✗ |
| Array Movement | — | — | ✓ | — |
| Feed Movement | — | — | ✓ | ✓ |

The array movement approach offers simple reconfigurability by using only a few actuators, which creates relatively low cost and less complex beam-steerable systems. These

features render this approach suitable for large arrays with many elements. Some of the TAs using array movement, e.g., Risley-prism-based beam-steerable antennas, often have limited frequency bandwidth that can be addressed by using true-time delays [61]. The feed movement is the simplest approach for mechanical reconfigurability as it requires adjusting light feed antennas as opposed to moving the large array apertures. Similar to array movement approaches, the RAs and TAs using feed movement often have limited bandwidth and efficiency. Therefore, the feed and array movements are appealing solutions for applications that require moderate speed, limited and affordable beam-steering over a narrow bandwidth.

## 7. Conclusions

We presented a review of different techniques for designing mechanically reconfigurable reflectarray and transmitarray antennas. We compiled and categorized the methodologies into three major groups and an additional hybrid group that combines one or more of these techniques. The antennas in each category feature specific properties, thus, offering a certain set of favorable performance metrics. We compared the scanning range, resolution and speed; we also compared the simplicity of implementation and operational frequency bandwidth. The techniques discussed herein can be applied to future mechanically reconfigurable antennas with superior performance, opening the door for new application spaces such as radars and communication systems.

**Author Contributions:** M.M., Z.Z., M.G., J.Z., M.M.H., J.H.B., and N.B. contributed to conceptualization, articulation, original draft preparation, review and the editing of the document. All authors have read and agreed to the published version of the manuscript.

**Funding:** This work relates to the Department of Navy awards N00014-19-1-2502 and N00014-19-1-2502 issued by the Office of Naval Research. The United States Government has royalty-free license throughout the world in all copyright-able material contained herein.

**Institutional Review Board Statement:** Not applicable.

**Informed Consent Statement:** Not applicable.

**Conflicts of Interest:** The authors declare no conflict of interest. The funders had no role in the design of the study; in the collection, analyses or interpretation of data; in the writing of the manuscript or in the decision to publish the results.

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
