# Peer review of "Mechanically Reconfigurable, Beam-Scanning Reflectarray and Transmitarray Antennas: A Review"

_applsci, doi:10.3390/app11156890_

Round 1

Reviewer 1 Report

The subject of the review is of great interest to antenna design engineers. A large amount of factual material of different techniques for designing mechanically reconfigurable antennas has been collected and analyzed.The work deserves publication after rather slight revision.

  • In conclusion, the summary should be drawn: what antenna categories are preferable for certain applications. Are the limitations in their applications fundamental, or one can hope  to overcome them as progress in this area of research activity. It should be added the short outlook to the last section: what directions the authors see for future progress?;
  • The illustrations should be improved.The actual experimental results are given only in Fig. 9b (the rest of the figures are conceptual in nature). It is not specified what conditions the various curves in Fig 9b.The location on one figure of a large number of chaotically oscillating curves does not allow the reader to identify them with confidence.It should be shown the scale at photo in Fig. 10b.

The subject of the paper corresponds to the profile of journal.

Author Response

The subject of the review is of great interest to antenna design engineers. A large amount of factual material of different techniques for designing mechanically reconfigurable antennas has been collected and analyzed. The work deserves publication after rather slight revision.

Response:

We would like to thank Reviewer #1 for their appreciation of this work and favorable comments regarding its content and originality. We have revised the manuscript, considering your comments. Please find below clear indications about the changes incorporated into the revised paper in response to your comments.

  1. In conclusion, the summary should be drawn: what antenna categories are preferable for certain applications. Are the limitations in their applications fundamental, or one can hope to overcome them as progress in this area of research activity. It should be added the short outlook to the last section: what directions the authors see for future progress?

Response:

We have added a new section named “Comparison and Discussion” and compared the performances of the three major approaches in this section. We have mentioned some specific applications that each approach is appropriate for. The challenges in each approach are discussed and some solutions are proposed.

Modifications into the revised paper:

  • A new section has been added.
  • A table has been added.
  • Some references have been added.

  1. The illustrations should be improved. The actual experimental results are given only in Fig. 9b (the rest of the figures are conceptual in nature). It is not specified what conditions the various curves in Fig 9b. The location on one figure of a large number of chaotically oscillating curves does not allow the reader to identify them with confidence. It should be shown the scale at photo in Fig. 10b.

Response:

This figure is taken from Reference 55, and the authors have not mentioned details about this figure. However, the feeds are displaced laterally along the X direction, and they are pointing out toward the center of the aperture. The authors have not revealed how much they displaced the feed for each different scanned gain patterns. Also, since we have taken Fig. 9(b) and Fig. 10(b) from their original sources, it is not possible for us to change the scale of the figures or to differentiate the curves more.

Modifications into the revised paper:

  • No change is applied.

Reviewer 2 Report

The paper provides an exhaustive and interesting review of mechanically reconfigurable beam-scanning reflectarray and transmitarray antennas. The manuscript is well written and accessible to non-specialists.

To further improve the quality of the paper, I would recommend including a final section comparing the different techniques for mechanical beam steering (advantages and drawbacks) and summarizing the main challenges and most promising developments on this field.

Some minor comments concerning typos:

-In page 5, in the caption of Fig. 3, it should be “micromotor” instead of “micrometer”.

-In page 9, line 298, it should be “was” instead of “wass”.

-Also in page 9, line 302, it should be “angle” instead of “angel”.

-References 47 and 50 are duplicated, and the same applies to references 51 and 56. Check the references to avoid more duplications.

Author Response

The paper provides an exhaustive and interesting review of mechanically reconfigurable beam-scanning reflectarray and transmitarray antennas. The manuscript is well written and accessible to non-specialists.

Response:

We would like to thank Reviewer #2 for their thorough evaluation and favorable inputs. Their constructive comments helped us to improve the clarity of the manuscript. Clarifications according to the comments of Reviewer #2 are listed below and discussed in detail.

  1. To further improve the quality of the paper, I would recommend including a final section comparing the different techniques for mechanical beam steering (advantages and drawbacks) and summarizing the main challenges and most promising developments on this field.

Response:

We have added a new section named “Comparison and Discussion” and compared the performances of the three major approaches in this section. We have mentioned some specific applications that each approach is appropriate for. The challenges in each approach are discussed and some solutions are proposed.

Modifications into the revised paper:

  • A new section has been added.
  • A table has been added.
  • Some references have been added.

  1. Some minor comments concerning typos: In page 5, in the caption of Fig. 3, it should be “micromotor” instead of “micrometer”.

Modifications into the revised paper:

  • The typo has been corrected.

  1. In page 9, line 298, it should be “was” instead of “wass”

Modifications into the revised paper:

  • The typo has been corrected.

  1. Also in page 9, line 302, it should be “angle” instead of “angel”.

Modifications into the revised paper:

  • The typo has been corrected.

  1. References 47 and 50 are duplicated, and the same applies to references 51 and 56. Check the references to avoid more duplications.

Modifications into the revised paper:

  • The redundant reference has been removed.
